# Contemporary HIV-1 envelope pseudovirus panels for detecting and assessing B cell lineages with broadly neutralizing antibody potential

Bette Korber[1]*, Michael S. Seaman[2], Nonhlanhla N. Mkhize[3,4], Kelli Greene[5], Hongmei Gao[5], Xiaoying Shen[5], Elizabeth Domin[5], Haili Tang[5], James Theiler[1], Kshitij Wagh[6], Penny L. Moore[3,4], Carolyn Williamson[7,8], James I. Mullins[9], Nicole A. Doria-Rose[10], David Montefiori[5], Elena E. Giorgi[11]*

1 New Mexico Consortium, Los Alamos, New Mexico, United States of America, 2 Center for Virology and Vaccine Research, Beth Israel Deaconess Medical Center, Harvard Medical School, Boston, Massachusetts, United States of America, 3 SAMRC Antibody Immunity Research Unit, Faculty of Health Sciences, University of the Witwatersrand, Johannesburg, South Africa, 4 National Institute for Communicable Diseases of the National Health Laboratory Service, Johannesburg, South Africa, 5 Department of Surgery, Duke University, Durham, North Carolina, United States of America, 6 Duke Human Vaccine Institute and Department of Medicine, Duke University School of Medicine, Durham, North Carolina, United States of America, 7 Institute for Infectious Diseases and Molecular Medicine, Division of Medical Virology, Faculty of Health Sciences, University of Cape Town, Cape Town, South Africa, 8 Centre for the AIDS Programme of Research in South Africa (CAPRISA), University of KwaZulu Natal, Durban, South Africa, 9 Department of Microbiology, University of Washington, Seattle, Washington, United States of America, 10 Vaccine Research Center, National Institute of Allergy and Infectious Diseases, National Institutes of Health, Bethesda, Maryland, United States of America, 11 Vaccine and Infectious Disease Division, Fred Hutchinson Cancer Center, Seattle, Washington, United States of America

* egiorgi@fredhutch.org (EEG); btk@lanl.gov (BK)

## Abstract

Although a protective HIV-1 vaccine has not yet been realized, significant progress has been made in vaccine designs that trigger B cell lineages with potential to produce broadly neutralizing antibodies (bnAbs). Advancing these strategies by optimizing vaccine boosting regimens requires early detection of maturing antibodies with neutralizing activity against native envelope glycoprotein (Env) trimers and streamlined strategies to identify antibodies as they begin to manifest desired levels of breadth and potency. Thus, we designed three types of pseudovirus screening panels based on Envs of contemporary HIV-1 isolates to facilitate detection of bnAb lineages that are on favorable trajectories during a vaccination course. The panels were selected from Tier 2 Transmitted Founder Lineage (TFL) HIV-1 Envs from placebo participants in the Antibody Mediated Prevention (AMP) efficacy trials. Using 15 bnAbs to evaluate the neutralization sensitivity of the viruses, we selected 8-member bnAb class-specific panels most sensitive to bnAbs representing their class: V2-apex, V3-glycan, CD4-receptor binding site (CD4bs), Membrane-Proximal External Region (MPER), or fusion peptide (FP). Next, we combined the most sensitive viruses among the class-specific panels to create a 12-virus panel to enable optimal detection of

**Data availability statement:** All neutralization data, aligned sequences, and virus panels are provided in supplementary data tables and are also available at the Harvard Dataverse Repository: dataverse.harvard.edu/dataset. xhtml?persistentId=doi:10.7910/DVN/3AVJUQ All env sequences in this study are available from GenBank, and accession numbers and metadata are listed in S8 Table. Additional information on neutralization assays and supporting protocols may be found at: http://www.hiv.lanl. gov/content/nab-reference-strains/html/home. htm.

**Funding:** This study was supported in part by two Collaboration for AIDS Vaccine Discovery (CAVD) grants from the Gates Foundation (INV-007368 and INV-036842 to DM) and Gates Foundation grant INV-048656 to Dan Barouch and BK, and in part by the Intramural Research Program of the National Institutes of Health (NIH). Overall support for the HIV Prevention Trials Network (HPTN) was provided by the National Institute of Allergy and Infectious Diseases (NIAID), Office of the Director (OD), National Institutes of Health (NIH), National Institute on Drug Abuse (NIDA), the National Institute of Mental Health (NIMH), and the Eunice Kennedy Shriver National Institute of Child Health and Human Development (NICHD) under Award Numbers UM1 AI068619 (HPTN Leadership and Operations Center), UM1 AI068617 (HPTN Statistical and Data Management Center), and UM1 AI068613 (HPTN Laboratory Center). Support for the HIV Vaccine Trials Network (HVTN) was provided by NIAID-NIH under Award Numbers UM1 AI068614 (HVTN Leadership and Operations Center), UM1 AI068635 (HVTN Statistical Data and Management Center (SDMC) and UM1 AI068618 (HVTN Laboratory Center). The contributions of the NIH author are considered Works of the United States Government. The findings and conclusions presented in this paper are those of the authors and do not necessarily reflect the views of the NIH or the U.S. Department of Health and Human Services. The funders had no role in study design, data collection and analysis, decision to publish, or preparation of the manuscript.

**Competing interests:** The authors have declared that no competing interests exist.

low-titer bnAb activity across epitope specificities. Finally, as HIV-1 continues to evolve greater levels of antigenic diversity and as current global pseudoviruses bnAb panels rely on viruses collected more than twenty years ago, we showed the importance of using contemporary viral panels to assess bnAb breadth and potency and designed a 12-virus panel representative of the spectrum neutralization profiles among AMP placebo viruses. We characterized pseudoviruses bearing each selected Env using standardized human sera to confirm their Tier 2 status and biological relevance. These updated panels enable sensitive screening of neutralization activity in vaccine studies and can also provide a realistic assessment of the expected breadth and potency of maturing responses against contemporary HIV-1 Envs.

## Author summary

A primary goal of HIV-1 vaccine research is to elicit neutralizing antibodies that can prevent infection, but HIV-1 is highly variable and stimulating effective responses against the diverse array of HIV-1 variants remains an unsolved challenge. However, the HIV-1 vaccine field has made significant progress in stimulating B cell lineages that produce antibodies known to have features able to ultimately generate potent broadly neutralizing antibodies. Inducing such precursor B cells is just the first step, as B cells need evolve in response to an antigenic stimulus through a process called affinity maturation to acquire potency and breadth against the natural diversity of circulating HIV-1 strains. The next step is to discover and optimize vaccine strategies that will selectively induce such maturation. To facilitate this work, we have designed panels of biologically relevant, contemporary HIV-1 Envelope pseudotyped viruses that will enable sensitive detection of neutralizing antibodies in human and animal vaccine studies. We also have defined a small screening panel intended to provide a quick but realistic assessment of the potential of a vaccine-stimulated antibody response or a monoclonal antibody being evaluated for clinical use to effectively counter and block currently circulating HIV-1 variants.

## Introduction

Defining the neutralizing breadth and potency of either HIV-1 monoclonal antibodies intended for clinical use or vaccine-elicited antibody responses is critical for understanding the potential efficacy of such interventions, but it is a complex and dynamic measure. HIV-1 continues to evolve ever-greater levels of genetic and antigenic diversity, showing increasing resistance to broadly neutralizing antibodies (bnAbs) at the population level over time [1–5], making it a necessity to continuously reassess the breadth and potency of bnAbs in circulating virus populations. However, pseudovirus panels expressing an array of HIV-1 envelope glycoproteins (Env-pseudotyped viruses) take a great effort to assemble, and so panels currently used to assess bnAb

breadth and potency often rely on viruses that were sampled 25–35 years ago. For example, Fig 1 highlights a collection of over 200 HIV Env pseudoviruses [7] that is still commonly used [8–10] and has historically been a valuable resource to the field but is diminishing in value over time due to its limitations for assessing relevant levels of breadth and potency given contemporary viral diversity. Based on data assembled in the Los Alamos HIV-1 database [7], the viruses in the panel were sampled during an older epoch in the HIV pandemic, between 1983 and 2008 (Fig 1A), when the circulating viruses were significantly less antigenically divergent [1–3,5]. Furthermore, this large viral panel includes some Tier 1 viruses (Fig 1B). Tier 1A viruses are very sensitive to neutralization, and Tier 1B represent viruses which tend to be on the more neutralization sensitive end of the spectrum [11]. Antibodies with a virus neutralization capacity restricted to Tier 1 viruses are readily elicited by vaccination, but they are not protective against new infections [12,13], thus the ease of neutralization of these viruses when they are included in test panels can be misleading if not carefully accounted for. In contrast, an antibody's ability to neutralize a diverse array of the more resistant Tier 2 viruses is a more biologically relevant measure for assessing the potential value of a neutralizing antibody response, as the preventing sexual HIV-1 transmission will likely require an ability to neutralize Tier 2 viruses [11]. Tier 3 viruses are the most resistant to neutralization, and these viruses are relatively rare by definition [14]. An additional issue for older panels is that some of the included pseudoviruses were derived from Env sequences obtained during chronic infection and based on bulk PCR amplification of a complex quasispecies, which can result in the technical artifact of in vitro recombination during amplification [15,16]. Such artificial recombinant sequences may not accurately represent a virus of natural origin [17,18].

To enable the selection of more biologically relevant panels for detecting promising neutralizing antibody responses in vaccines studies, we utilized transmitted-founder viruses isolated from the placebo group participants in the Antibody Mediated Prevention (AMP) trials. The AMP trials were designed to test whether passive infusion every 8 weeks of the CD4-binding site (CD4bs) bnAb VRC01 could prevent HIV-1 infection [19]. The reason we restricted our panel selection to the placebo group was to avoid the possibility of selection by VRC01. Two AMP clinical trials of at-risk individuals were conducted in parallel: HVTN 703/HPTN 081 including women in sub-Saharan Africa in regions with predominantly HIV-1 C clade infections, and HVTN 704/HPTN 085 including cisgender men and transgender persons in the Americas and Europe with predominantly B clade infections, although also including some F clade and B/F recombinant viruses. VRC01 infusions did not significantly reduce HIV-1 acquisition [19]. However, infection by VRC01-sensitive viruses was reduced [20], and high neutralization titers against the infecting virus were associated with reduced viral loads in breakthrough cases [21].

Here we define strategically selected panels composed of these well-characterized AMP placebo group viruses for assessing progress in HIV-1 vaccine development, enabling the creation of valuable sets of biologically relevant and relatively contemporary Tier 2 pseudoviruses [14]. In large longitudinal vaccine trials, it is critically important to be able to detect neutralizing responses in vaccinees just as they are beginning to make promising neutralizing responses. Most of the HIV-1 neutralizing antibody vaccine efforts coalesce around induction of classes of bnAbs targeting one or more of the select Env epitope regions [22]. We reasoned that identifying small panels of Tier 2 pseudoviruses that were the most neutralization sensitive for specific classes of bnAbs would enable detection of early signs of heterologous neutralizing breadth during a vaccination course and facilitate monitoring progress in vaccine studies in real time. For example, a strategy of keen current interest is the design of vaccine immunogens that specifically prime rare precursors of bnAb lineages that can perhaps be directed towards mature bnAb responses [22–26] through a series of heterologous boosts. One promising boosting strategy first tracks the natural development of breadth of a potent bnAb *in vivo*, then defines the improbable mutations in the antibody lineage essential for expanding breadth to enable the design of boosting immunogens that can specifically select for those mutations [22,27,28]. These and other approaches are being explored in different vaccine strategies intended to trigger antibodies against multiple epitope targets, including CD4bs, V2-apex, V3-glycan, fusion peptide (FP), and the membrane proximal external region (MPER). Thus, panels of sensitive viruses are critically needed for detecting early responses that are still weak and rare in sera to compare vaccine strategies under

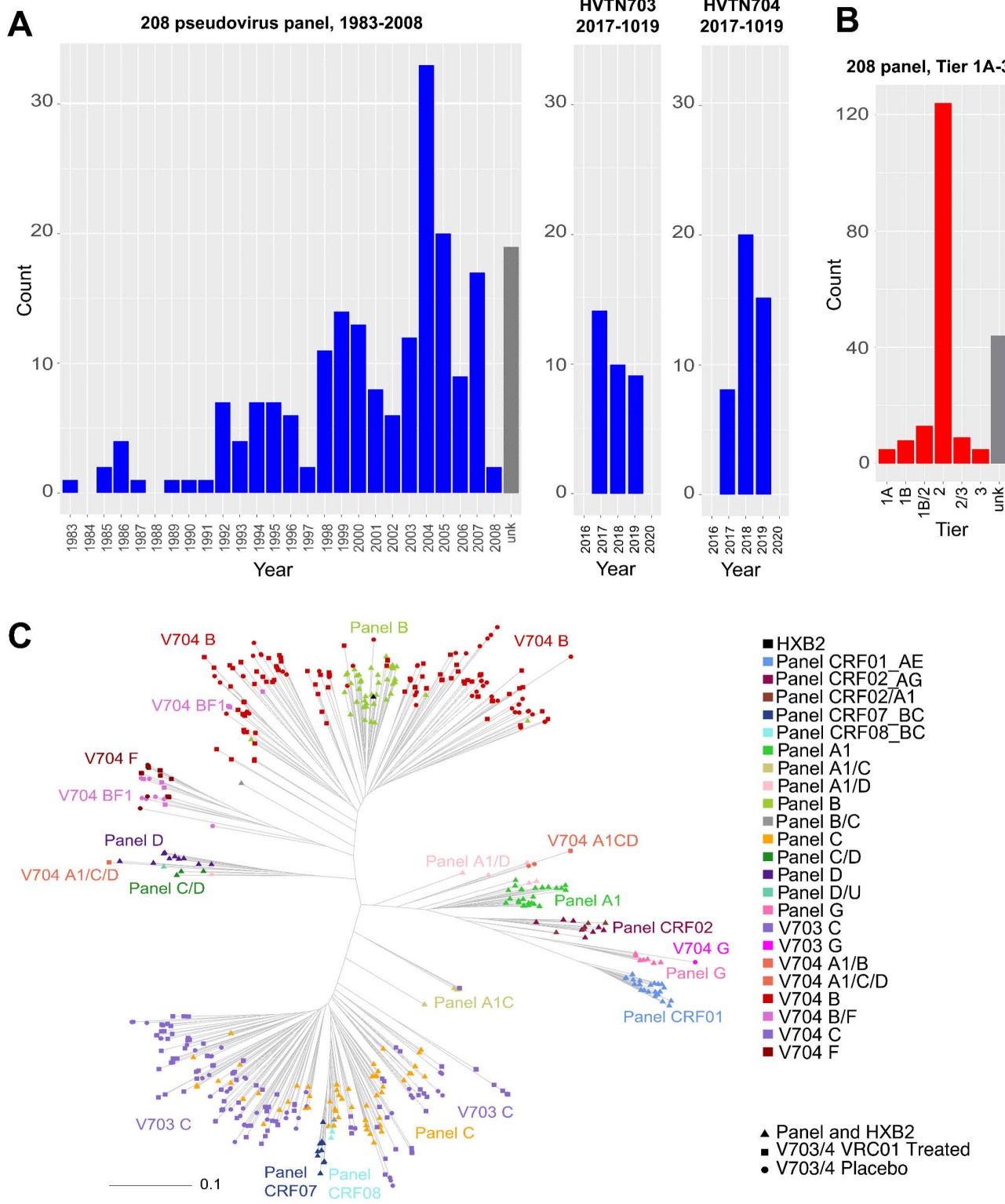

**Fig 1. Characterization of the 208 virus panel often used to explore breadth and potency of HIV-1 bnAbs. (A)** Sampling years of the 208-virus panel [6] compared to the AMP placebo viruses. **(B)** Tier profile of the 208-virus panel. In contrast, the AMP placebo viruses that were down-selected for reference panels in this study were all demonstrated to be Tier 2 as a criterion for inclusion. **(C)** Mid-point rooted phylogenetic tree from a version of the

208 virus panel based on data that was retrieved using the CATNAP tool from the Los Alamos HIV-1 Database accessed on July 1, 2025 [7] merged with 345 Env sequences sampled from the AMP trials. The AMP viruses are much more diverse in terms of longer branch lengths within both B and C clades, and particularly within the B clade only a narrow sublineage was sampled in the older panel. The panel of 208 viruses includes many additional globally circulating clades relative to the AMP trial viruses, although multiple F clade and BF recombinants were sampled in Peru during the AMP 704 trial, and these lineages were not present in the panel of 208 viruses. The panel of 208 viruses continues to be a core reference panel for the HIV field and is still commonly used to characterize antibodies of key importance in recent studies [8–10]. Note that the exact composition of the 208 panel can vary by a few viruses between studies. The version used in this tree included 205 viruses that were available in the CATNAP tool at the Los Alamos database.

development and build on those with the most promise. Equally important is detecting antibodies that are beginning to acquire levels of breadth and potency such that they have potential for application for clinical use given high level of natural diversity found in circulating HIV-1. For this second use, we used k-means clustering as an approach to down-select twelve viruses, a practical size for screening responses in large studies, that reflected the spectrum of antigenic diversity found in the full set of AMP placebo group of viruses. This panel may also be helpful for assessing the breadth and potency of monoclonal antibodies being studied for clinical applications for prevention or therapeutics.

## Results

### Comparing AMP viral sequences to historically commonly used neutralizing antibody evaluation panels

The Transmitted Founder Lineage (TFL) viruses [2,17] collected in the AMP studies provide a valuable resource to identify sets of biologically relevant Env-pseudotyped viruses to enable characterization of neutralizing antibody responses. Fifteen antibodies were included to represent different classes of bnAbs. The "LS" in many of the antibody names refers to amino acid changes M428L and N434S introduced into the constant fragment crystallizable (Fc) region of IgG regions that can improve pharmacokinetic profiles in vivo for antibodies being considered as infection-prevention and/or therapeutic agents [29]. To represent the diversity of viral sequences sampled in the AMP studies and compare that with older panels for phylogenetic analysis, we used the full set of viral sequences described in Mkhize et al. [2]. For panel selection and to compare neutralization responses, however, we focused on just the TFL viruses from individuals in the AMP placebo groups to ensure exclusion of the AMP viruses from the VRC01 treatment group that might be impacted by VRC01-imposed selection [20]. The full dataset included 118 pseudoviruses derived from 76 placebo individuals The neutralization data can be found in S1 Table, and viral nomenclature in S2 Table. Note that often more than one TFL virus was identified in a single individual. There were 33 individuals included from the HVTN 703/HPTN 081 trial and 43 from the HVTN 704/HPTN 085 trial. AMP viral sequences were sampled using laboratory protocols designed to accurately reflect within-host quasispecies diversity and to maintain the integrity of the natural sequences [18].

Among the B and C clade variants that were the most common circulating clades in the geographic regions where the AMP trials were conducted, the diversity of the older panel viruses is markedly less than the diversity of the AMP variants (Figs 1, S1 and S2). These older HIV-1 pseudovirus panels are impacted to varying degrees by the three issues discussed in the introduction: (i) age (Fig 1A), (ii) Tier status (Fig 1B), and (iii) in vitro recombination [15,16]. Phylogenetic analysis shows that the genetic diversity found within the B and C clades in the commonly used 208 viral panel previously described in Doria-Rose et al. [6] is substantially less than what is found among the more contemporary AMP viruses (Fig 1C). With respect to clade distributions, Fig 1C also highlights a limitation of the AMP viruses, which is that many clades that are important globally are not represented at all in the AMP trial, e.g., the A clade. As a counterpoint to this issue, the F clade is common in the South American HIV-1 epidemic, and F and BF recombinants were not included among the 208 panel, but they were relatively frequently sampled in the AMP study. Other panels that are currently used to explore neutralization breadth also show substantially reduced genetic diversity relative to the AMP viruses. S1 Fig shows reduced diversity in a global panel of 119 viruses that is also often used to assess breadth and potency [30–33] relative to AMP B and C clade viruses. Sixty-eight percent of the viruses in this 119 panel are also found in the 208 panel

and represent an even more restricted sampling of B and C clade diversity. S2 Fig shows a panel of 200 C clade viruses, focusing on just the C clade because it is the most common clade circulating in southern Africa where HIV-1 is most highly prevalent [3]; 15% of the viruses in this C clade panel are shared with the 208 panel. Overall, this panel is also less diverse than the more contemporary C clade viruses sampled in the AMP HVTN 703 trial, although it is larger and more recently sampled, so better captures the spectrum of contemporary diversity of the C clade as compared to the two global panels shown in Figs 1C and S1.

The increasing genetic diversity of the Env protein over time has manifested as increasing bnAb resistance. We illustrate this relationship by using two summary statistics that reflect bnAb breadth (Fig 2A and 2B) and potency (Fig 2C). Thirteen of the fifteen antibodies used in this study to characterize the neutralization profiles of AMP placebo viruses (the AMP virus IC50 and IC80 neutralization data is included in S1 Table) were also characterized using the full set of viruses in the 208 panel with data readily available through the HIV-1 Immunology database tool CATNAP (**C**ompile, **A**nalyze and **T**ally **Na**b **P**anels) [7], allowing us to directly compare the two datasets. In the AMP trials, the CD4bs bnAb VRC01 showed a significant protective effect against viruses with an IC80 < 1 µg/ml, providing a useful benchmark for predicting the potential for a given bnAb to confer biologically meaningful protection against diverse circulating viral variants [20]. Thus, we calculated the fraction of viruses in the 208 panel that had an IC80 of <1 µg/ml and compared it to the fraction found among the AMP placebo viruses for each of 13 bnAbs available for comparison. For 13 out of the 13 bnAbs compared, the more contemporary AMP viruses had fewer sensitive viruses with an IC80 of <1 µg/ml than did the 208 panel (Fig 2A), with the Wilcoxon signed-rank test p-value of 0.0002 rejecting the null hypothesis that frequency of sensitive viruses for the 13 bnAbs was comparable in the two data sets. The reduction in the number of viruses sensitivity to V2 apex and FP bnAbs among the AMP placebo viruses relative to the 208 panel was particularly concerning. Because C clade viruses tend to be more sensitive to V2 apex bnAbs than B clade [34], we restricted the analysis to just the C clade viruses in the 208 panel and among AMP placebo viruses (Fig 2B). In a C clade only setting, again 13 out of 13 viruses bnAbs had fewer sensitive viruses with an IC80 of <1 µg/ml in the AMP placebo group, but the V2 apex bnAbs have higher fractions of IC80 of <1 µg/ml among the C clade viruses, indicating that V2 apex bnAbs may be more useful in regions where the C clade dominates the epidemic (Fig 2B). Finally, using an inclusive strategy to detect positive neutralization responses and considering all cases of detectable neutralization activity at the highest level of antibody used to determine the IC50s values, the *potency* of the neutralization responses was diminished among the AMP viruses relative to the 208 panel, again for all 13 antibodies tested (Fig 2C). The relatively high potency of the CAP256-VRC26.25 LS compared to other bnAbs may make this antibody particularly valuable in geographic regions with predominantly C clade epidemics (Fig 2C), however this is not the only consideration as CAP256-VRC26.25 LS activity as found to decline in vivo after infusion more rapidly than expected given serum antibody concentrations [35].

## SHEP-T2: A panel of pseudoviruses for detecting early neutralizing responses

We selected a panel of 12 pseudoviruses from AMP placebo recipients to facilitate detection of low titers of vaccine-elicited bnAbs in early stages of development. This panel is intended for detection of the capacity to neutralize heterologous Envs to measure the progress in *shepherding* of vaccine-stimulated antibody lineages towards greater neutralizing breadth and potency. We call this the Sensitive HIV-1 Envelope Panel, Tier 2 (SHEP-T2). Panel inclusion was based on IC50 profiles of 15 bnAbs representing 5 distinct antibody classes: CD4bs, V2 apex, V3 glycan, FP, and MPER, and so it is designed to have the capacity to detect early responses across antibody classes (Fig 3). An example of the utility of this panel would be to assess different boosting strategies intended to advance lineages toward heterologous virus neutralization and the beginnings of neutralization breadth [22,27,28] after vaccine priming [22–26].

To select SHEP-T2, we first ranked viruses by their sensitivity across bnAbs and then imposed a requirement for the inclusion of the two most sensitive viruses to each antibody tested in the panel of 12 (Fig 3). Our ranking strategy required a single virus to represent each infected individual, although multiple variants established the infection in some cases and

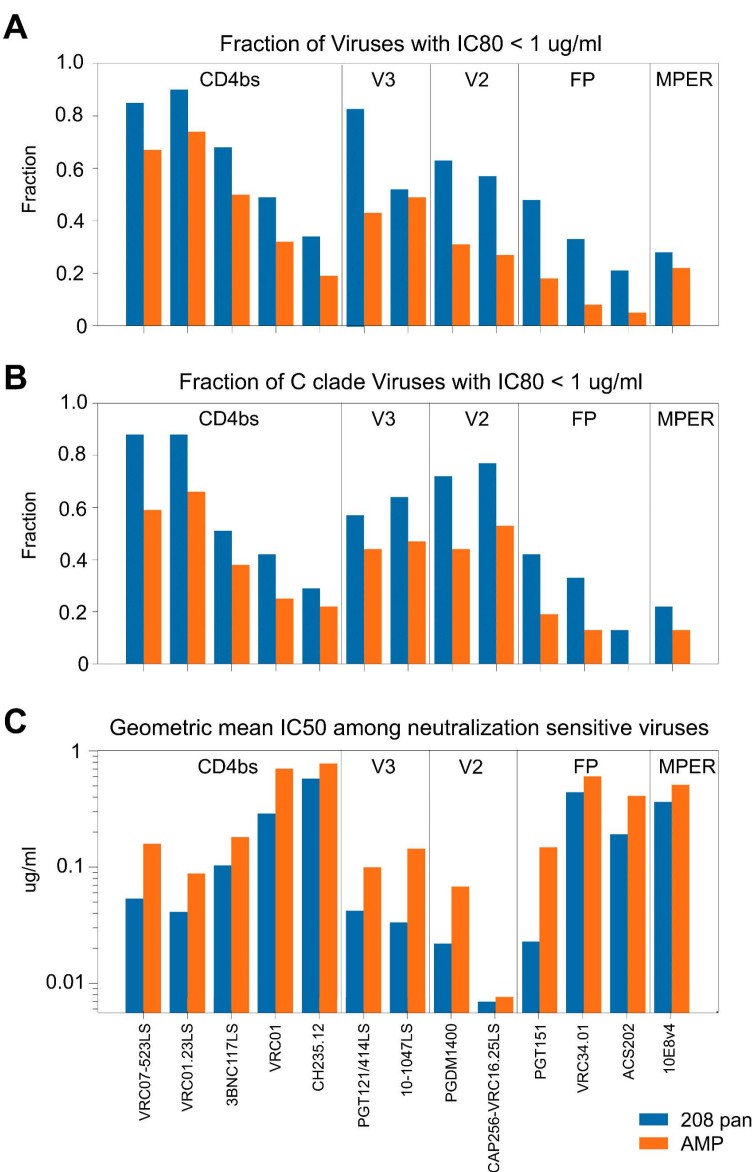

**Fig 2. Summary statistics of bnAb breadth and potency comparing the 208 viral panel and the AMP virus placebo set. (A)** The fraction of viruses in the 208 panel (blue) versus the AMP panel (orange) that achieved IC80 of <1 ug/ml, the potency threshold that was found to be protective for VRC01 in the AMP trials [20]. **(B)** The fraction of the viruses that achieved IC80 of <1 ug/ml restricted to include only the C clade viruses in AMP or the 208 virus panel. The C clade viruses in the 208 panel included all C clade viruses as well as circulating recombinant forms CRF07 and CRF08, which are essentially C clade in Env and are commonly found in parts of Asia. **(C)** The geometric mean IC50 titer among all positive tests for either the AMP or the 208 virus panel. For IC50 scores, smaller numbers are indicative of less antibody required to achieve a 50% inhibition and as such reflect more potent responses. This figure includes comparisons of each of the 13/15 antibodies in this study that had complete testing available through the CAT-NAP tool for the 208 panel viruses [7] enabling a direct comparison with the AMP placebo viruses.

more than one TFL was tested. Twenty-nine such individuals among the 76 AMP placebo participants had multiple viruses sampled and evaluated for bnAb sensitivity. BnAb sensitivity profiles of pseudoviruses from the same individual tended to be similar (S1 Table), and the most sensitive pseudovirus from within each person was chosen as representative, as our purpose with this panel was to *sensitively* detect early traces of heterologous serum neutralization in vaccine studies.

| panel number | Sequence Name | Geometric mean of IC50 Rank | Rank | CD4bs | | | | | | V3 | | V2 | | FP | | | MPER | clade |
|---|---|---|---|---|---|---|---|---|---|---|---|---|---|---|---|---|---|---|
| | | | | 1-18 | VRC07-523 LS | VRC01-23 LS | 3BNC117 LS | VRC01 | CH235.12 | PGT121.414 LS | 10-1074 LS | PGDM 1400 | CAP256-VRC26.25 LS | PGT151 | VRC34.01 | ACS202 | 10E8v4 | |
| 1 | V704_0128_220_RE_NT_pblib001_52 | 10.21 | 1 | 0.023 | 0.006 | 0.011 | 0.026 | 0.040 | 0.021 | 0.018 | 0.059 | 0.008 | 0.002 | >25 | 0.190 | 22.918 | 1.221 | F/B |
| 2 | V704_0855_080_RE_NT_pblib001_18 | 11.47 | 2 | 0.012 | 0.003 | 0.002 | 0.009 | 0.013 | >25 | 0.050 | 0.061 | >25 | >25 | 0.007 | >25 | >25 | 0.134 | B |
| 3 | V704_1180_070_RE_NT_pblib001_87 | 12.59 | 3 | 0.005 | 0.045 | 0.013 | 0.016 | 0.170 | 0.145 | >25 | >25 | 0.188 | >25 | 0.009 | 0.280 | 0.273 | 0.320 | A/B |
| 4 | V703_0472_030_RE_con_s | 13.75 | 4 | 0.005 | 0.010 | 0.005 | >25 | 0.021 | 0.050 | >25 | 7.940 | 0.100 | 0.010 | 0.028 | 5.670 | >10 | 0.420 | C |
| 5 | V703_2304_150_RE_con_s | 13.77 | 5 | 0.060 | 0.020 | 0.011 | 0.060 | 0.114 | 0.080 | 0.070 | 0.060 | 0.010 | 0.010 | >10 | 0.030 | >10 | 0.360 | C |
| 6 | V703_2805_080_RE_con_s | 15.12 | 6 | 0.027 | 0.010 | 0.029 | 0.020 | 0.267 | 0.240 | >25 | >25 | 0.002 | 0.000 | 0.060 | 0.230 | >10 | 1.220 | C |
| 7 | V704_1535_030_RE_NT_pblib001_47 | 15.30 | 7 | 0.052 | 0.068 | 0.018 | 0.019 | 0.088 | >25 | 0.011 | 0.071 | 1.109 | >25 | 0.197 | 0.390 | 0.512 | 0.015 | B |
| 8 | V704_1528_240_RE_NT_pblib001_17 | 15.69 | 9 | 0.060 | 0.439 | 0.114 | 0.092 | 2.390 | 7.107 | 0.003 | 0.004 | >25 | >25 | 1.418 | 0.040 | 0.041 | 0.243 | B |
| 9 | V704_0907_130_RE_NT_pblib001_32 | 16.01 | 10 | 0.016 | 0.193 | 0.019 | 0.199 | 0.364 | 15.857 | 0.002 | 0.006 | >25 | >25 | 0.017 | 0.>25 | 0.236 | 0.655 | B |
| 10 | V703_1313_040_RE_con_s | 17.08 | 12 | 0.046 | 0.040 | 0.070 | 0.110 | 0.686 | >10 | 0.010 | 0.060 | 0.004 | 0.002 | >10 | >10 | >10 | 0.010 | C |
| 11 | V703_2117_110_RE_sga2A10_s | 20.38 | 17 | 0.170 | 0.040 | 0.060 | 0.110 | 0.229 | 0.520 | >25 | >25 | 0.001 | 0.000 | >10 | 0.810 | >10 | 0.880 | C |
| 12 | V704_0445_180_RE_NT_pblib001_22 | 26.09 | 25 | 0.012 | 0.063 | 0.021 | >25 | 82.985 | 0.205 | 0.291 | 0.086 | 0.285 | >25 | 0.083 | >25 | 0.080 | 1.490 | B |
| na | V703_0537_110_RE_sga4H1_s | 15.32 | 8 | 0.010 | 0.010 | 0.005 | 0.020 | 0.090 | 1.140 | 0.010 | 0.080 | 0.090 | 0.010 | >10 | >10 | >10 | 0.680 | C |
| na | V704_2684_181_RE_NT_pblib001_13 | 16.32 | 11 | 0.010 | 0.044 | 0.012 | 0.051 | 0.090 | >25 | 0.006 | 0.019 | >25 | >25 | >25 | >25 | 0.913 | 0.030 | B |

**Clinical trial**

| HVTN 704/HPTV 085 | |
|---|---|
| HVTN 703/HPTN 081 | |

**IC50**

| HVTN 703: >10 | |
|---|---|
| HVTN 704: >25 | |
| HVTN 704: >10 - 25 | |
| >1 - 10 | |
| >0.1 - 1 | |
| >0.01 - 0.1 | |
| >0.001 - 0.01 | |
| <= 0.001 | |

**Fig 3. IC50 scores and ranks for the SHEP-T2 viral panel.** The color key represents the *IC50 values* in a logarithmic scale. We computed the geometric average of the rank of the IC50 score across all bnAbs tested, such that lower values indicate greater sensitivity across all antibodies; this was used as summary statistic to compare the overall sensitivity of the AMP pseudoviruses (S3 Table). This sensitivity ranking was used to guide the selection of the SHEP-T2 panel, with the caveat the panel was required to include at least two of the three most sensitive viruses for each bnAb. White font with an underscore indicates that the IC50 value was one of the 3 lowest observed for an antibody among all the AMP placebo viruses. Replacing the 8th- and 11th-ranked pseudoviruses with the 17th- and 25th enabled us to meet this second selection criterion to arrive at the final panel of 12 highly sensitive Tier 2 TFL viruses for screening purposes (S3 Table).

Complete IC50 and IC80 neutralizing data are included in S1 Table, GenBank sequence accession numbers and viral nomenclature keys are available in S1 Table.

To rank the overall sensitivity of the viruses, the IC50 titers were replaced by their rank from most to least sensitive for each bnAb. We used the ranks rather than IC50 titers so that the moderately potent (yet useful) bnAbs would be comparably weighted to the few more highly potent bnAbs in selecting the SHEP-T2 panel. Pseudoviruses with identical IC50 titers for a given antibody were assigned the same average rank (*e.g.,* if IC50 titers are [0.01, 0.02, 0.02, 0.05], assigned ranks are [1, 2.5, 2.5, 4]). We then computed the geometric average of these ranks as a summary statistic for each pseudovirus (S3 Table). Using a geometric rather than arithmetic average gave more emphasis to lower ranks, favoring viruses that were the most sensitive to multiple antibodies tested. Finally, we required that the panel include at least two of the three most sensitive viruses for each bnAb. The first iteration of the panel did not include two of the three most sensitive viruses for the V2 apex bnAbs PGDM1400 and CAP256-VRC 2.6.25LS, and the FP antibody ACS202; replacing the 8th- and 11th-ranked pseudoviruses with the 17th- and 25th enabled us to meet this selection criterion for the final panel (Fig 3 and S3 Table).

## BnAb class-specific panels

SHEP-T2 was designed to facilitate detection of early responses to *any* of five key classes of antibodies, but often vaccine or monoclonal antibody studies are focused on a specific bnAb epitope. To facilitate early detection of heterologous breadth of each class of bnAbs, we also provide 5 panels of 8 Tier 2 pseudoviruses selected to be specifically sensitive to a particular class of bnAb (Fig 4). Germline precursors of antibody lineages that ultimately give rise to bnAbs have a

**CD4bs**

| Sequence Name | Rank full set | SHEP-T2 set | Rank, CD4bs only | 1-18 | VRC07-523 LS | VRC01-23 LS | 3BNC1 17 LS | VRC01 | CH235.12 | Clade |
|---|---|---|---|---|---|---|---|---|---|---|
| 1 V704_0855_080_RE_NT_pblib001_187 | 2 | 2 | 3.38 | 0.012 | 0.003 | 0.002 | 0.009 | 0.013 | >25 | B |
| 2 V704_0128_220_RE_NT_pblib001_52 | 1 | 1 | 3.39 | 0.023 | 0.006 | 0.011 | 0.026 | 0.04 | 0.021 | F1/B |
| 3 V703_0472_030_RE_con_s | 4 | 4 | 4.87 | 0.005 | 0.01 | 0.005 | >25 | 0.021 | 0.05 | C |
| 4 V704_1180_070_RE_NT_pblib001_87 | 3 | 3 | 5.00 | 0.005 | 0.045 | 0.013 | 0.016 | 0.17 | 0.145 | A1/B |
| 5 V704_1535_030_RE_NT_pblib001_471 | 7 | 7 | 8.95 | 0.052 | 0.068 | 0.018 | 0.019 | 0.088 | >25 | B |
| 6 V704_1775_030_RE_NT_pblib001_135 | 14 | | 8.35 | 0.014 | 0.029 | 0.031 | 0.109 | 0.09 | 0.05 | C |
| 7 V703_2805_080_RE_con_s | 6 | 6 | 9.87 | 0.027 | 0.01 | 0.0287 | 0.02 | 0.2668 | 0.24 | F1 |
| 8 V704_1783_150_RE_NT_pblib001_144 | 21 | | | 0.006 | 0.064 | 0.009 | 0.036 | 0.135 | >25 | B |

**FP**

| Seq Name | Rank full set | SHEP-T2 set | Rank, FP only | PGT151 | VRC34.01 | ACS202 | Clade |
|---|---|---|---|---|---|---|---|
| 1 V704_1528_240_RE_NT_pblib001_17 | 9 | 8 | 4.93 | 1.418 | 0.04 | 0.041 | B |
| 2 V703_0629_150_RE_pblib002_s | 23 | | 5.48 | 0.0032 | 0.16 | >10 | C |
| 3 V703_0566_160_RE_con_s | 15 | | 6.91 | 0.007 | 0.08 | >10 | C |
| 4 V704_1180_070_RE_NT_pblib001_87 | 3 | 3 | 7.73 | 0.009 | 0.28 | 0.273 | A1/B |
| 5 V704_0445_180_RE_NT_pblib001_22 | 25 | 12 | 14.96 | 0.083 | >25 | 0.08 | B |
| 6 V704_0575_060_RE_NT_pblib002_16 | 39 | | 8.00 | >25 | 0.24 | 0.039 | B |
| 7 V703_2304_150_RE_con_s | 5 | 5 | 11.41 | >10 | 0.03 | >10 | C |
| 8 V703_0646_051_RE_con_s | 42 | | 11.89 | 0.07 | 0.07 | >10 | C |

**V3**

| Seq Name | Rank full set | SHEP-T2 set | Rank, V3 only | PGT121.414 LS | 10-1074 LS | Clade |
|---|---|---|---|---|---|---|
| 1 V704_0907_130_RE_NT_pblib001_32 | 10 | 9 | 1.73 | 0.002 | 0.006 | B |
| 2 V704_1528_240_RE_NT_pblib001_17 | 9 | 8 | 2.00 | 0.003 | 0.004 | B |
| 3 V704_0847_030_RE_NT_pblib001_34 | 43 | | 2.74 | 0.01 | 0.003 | B |
| 4 V704_0026_231_RE_NT_pblib001_3 | 24 | | 5.29 | 0.006 | 0.017 | B |
| 5 V704_2684_181_RE_NT_pblib001_13 | 11 | | 5.92 | 0.006 | 0.019 | B |
| 6 V703_0566_160_RE_con_s | 15 | | 6.12 | 0.01 | 0.01 | C |
| 7 V704_2541_080_RE_NT_pblib001_13 | 13 | | 8.49 | 0.026 | 0.008 | B |
| 8 V703_1313_040_RE_con_s | 12 | 10 | 11.46 | 0.01 | 0.06 | C |

**V2**

| Seq Name | Rank full set | SHEP-T2 set | Rank, V2 only | PGDM1400 | CAP256-VRC26.25LS | Clade |
|---|---|---|---|---|---|---|
| 1 V703_2117_110_RE_sga2A10_s | 17 | 11 | 1.73 | 0.0007 | 0.0001 | C |
| 2 V703_2149_060_RE_sgaB10_s | 20 | | 2.45 | 0.0012 | 0.0001 | C |
| 3 V703_2018_240_RE_sga6A1_s | 32 | | 4.58 | 0.0031 | 0.0001 | C |
| 4 V703_2805_080_RE_con_s | 6 | 6 | 5.45 | 0.0019 | 0.0004 | C |
| 5 V703_1675_080_RE_con_s-modified | 16 | | 6.12 | 0.01 | 0.0001 | C |
| 6 V703_1313_040_RE_con_s | 12 | 10 | 9.38 | 0.004 | 0.0018 | C |
| 7 V704_0128_220_RE_NT_pblib001_52 | 1 | 1 | 10.39 | 0.008 | 0.002 | F1/B |
| 8 V704_1775_030_RE_NT_pblib001_135 | 14 | | 10.95 | 0.002 | 0.028 | F1 |
| Alternates | | | | | | |
| 9 V703_2304_150_RE_con_s | 1 | 5 | 13.72 | 0.01 | 0.01 | C |
| 10 V703_0646_051_RE_con_s | 42 | | 13.72 | 0.01 | 0.01 | C |

**MPER**

| Seq Name | Rank full set | SHEP-T2 set | Rank 10E8v4 | 10E8v4 | Clade |
|---|---|---|---|---|---|
| 1 V703_1313_040_RE_con_s | 12 | 10 | 1 | 0.01 | C |
| 2 V704_1535_030_RE_NT_pblib001_471 | 7 | 7 | 2 | 0.015 | B |
| 3 V704_2684_181_RE_NT_pblib001_13 | 11 | | 3 | 0.03 | B |
| 4 V704_0026_231_RE_NT_pblib001_3 | 24 | | 4 | 0.07 | B |
| 5 V703_0279_110_RE_con_s-modified | 51 | | 6 | 0.08 | C |
| 6 V704_0372_250_RE_NT_pblib001_7 | 50 | | 7 | 0.087 | B |
| 7 V704_0855_080_RE_NT_pblib001_187 | 2 | 2 | 8 | 0.134 | B |
| 8 H704_1835_150_RE_p001s_2484A | 37 | | 11 | 0.147 | B |

IC50
HVTN 703: >10
HVTN 704: >25
HVTN 704: >10 - 25
>1 - 10
>0.1 - 1
>0.01 - 0.1
>0.001 - 0.01
<= 0.001

Clinical trial
HVTN 704/HPTV 085
HVTN 703/HPTN 081

**Fig 4. IC50 scores for the eight most highly sensitive viruses among the AMP placebo viruses for each class of antibody studied.** As for SHEP-T2, viruses were ranked according to the geometric mean of the rank scores but restricted to class-specific antibodies. Here we required the inclusion of 3 of the most sensitive pseudoviruses for each antibody within the class. V2 antibodies have a primary panel that includes 8 highly sensitive

viruses. A glycan N130 can confer resistance to many important V2 antibodies, but not to the two apex PGDM1400 or CAP256 VRC16.25 LS studied here; in fact, the viruses most sensitive to these particular V2 apex antibodies were enriched for it, as the glycosylation site at N130 was present in 7/8 panel viruses. Because this panel was intended as a tool for screening for V2 apex antibodies early in their development, we included two more alternate viruses that were still quite sensitive to the V2 apex antibodies studied here but did not have the glycan at N130, making a 10 member panel rather than 8. The initial MPER and CD4bs panels each had a virus that ranked well in terms of sensitivity, but that did not grow well enough to enable distribution, so they were replaced. For the MPER panel V703_1060_080_RE_con_s was replaced with H704_1835_150_RE_p001s_2484A. For the CD4bs panel V703_0537_110_RE_sga4H1_s was replaced with V704_1783_150_RE_NT_pblib001_144.

very limited, if any, neutralizing capacity [22–26]; we confirmed this by finding no detectable IC50 titers for what are best estimates of 26 germline precursors against any of the 40 viruses included in the SHEP-T2 and class-specific antibody panels (S4 Table). Thus, this panel would not be expected to be sensitive to germline precursors, rather it will be useful for sensitive detection of early signs of affinity maturation toward biologically relevant breadth and potency.

Two issues merit special consideration that are relevant to the class of bnAbs that recognize the V2 apex region of Env. First, some but not all V2-targeting bnAbs are sensitive to the presence of a potential N-linked glycan at position N130; potential N-linked glycosylation sites are defined by the amino acid motif NX[S/T]. The two V2 bnAbs used in this study to help select the AMP panels, PGDM1400 and CAP256-VRC26.25 are not impacted by the presence of an N130 glycan, while it is associated with increased resistance to other V2 apex bnAbs like PG9, PG16, and CH01 [34], and it has been shown to play a key role in limiting breadth of a developing V2 specific antibody response [36]. Seven of the eight pseudoviruses most sensitive to the PGDM1400 and CAP256-VRC26.25 carry an N-linked glycosylation site at N130 (S3 Fig), and we felt it is important to have options in this panel that do not carry this site to enable sensitive detection of lineages which N130-glycan can inhibit. To address this, we selected from among other AMP Envs two additional alternative V2-sensitive panel members that do not have a glycosylation site at N130, and that we had already determined to be Tier 2 viruses that can readily be grown up for panel reagents, and that are still highly sensitive to PGDM1400 and CAP256-VRC26.25, but were just not among the top eight (Figs 4 and S3).

The other important issue with V2 apex bnAbs is that B clade viruses tend to be much more resistant to V2-targeting bnAbs compared to other clades [34]; consistent with past findings, the C clade and F clade viruses we studied in the AMP clinical trials were much more susceptible to V2 bnAbs than B clade viruses (S3 Fig and S5 Table). Because of the generally high level of resistance of B clade viruses to these V2 bnAbs, there are no B clade viruses included in the V2 sensitivity panel; we captured some diversity outside of the highly sensitive C clade by including an F1 clade and a BF1 recombinant virus which is F1-like, not B-like, in the V2 region.

To facilitate interpretation of the panel results for investigators who might incorporate them in future studies, for the bnAbs that were studied both in this paper and in the Bricault et al. paper [30], which defined sequence-based signatures of neutralization sensitivity, we created S3–S6 Figs. These figures overlay the signature sites identified in Bricault et al. onto the bnAb class specific sensitive sets versus other AMP placebo viruses to enable quick assessment of the amino acid composition of the sensitivity panels in key signature positions in the relevant epitope regions. Hypervariable regions within the variable regions of the HIV-1 Env rapidly evolve in vivo by insertion and deletion, and we have previously demonstrated that the length, net charge, and number of potential glycosylation sites in these regions can significantly impact bnAb escape [34]. Thus, we summarize specific Env variable loop characteristics associated with bnAb neutralization sensitivity/resistance for each panel (S3–S6 Figs) [34]. We also provide regional alignments that span the known epitope targeting regions for the sequences of the viruses included in each class-specific panel (S3–S6 Figs).

## Selecting a panel of 12 viruses to be representative of naturally circulating variants

Our aim for this panel was to define a relatively small panel of pseudoviruses that would capture the potential of an antibody response to interact with a spectrum of antigenically diverse phenotypes observed among the larger set of AMP

placebo viruses. We call this Representative Tier 2 panel REP-T2. For SHEP-T2 we used the most *sensitive* pseudovirus from each participant when multiple viruses had been sampled and tested, as we were designing panels meant to be highly sensitive while still biologically relevant. In contrast, we thought a better choice for a pseudovirus panel to assess breadth-potency at the population level would be the *least sensitive* variant from each host, as the AMP trial demonstrated that the presence of bnAbs can select against more sensitive forms of the virus, and we hoped to achieve a realistic perspective of the neutralizing potency of sera against circulating strains at the populations level [20,37]. Both the most and least bnAb sensitive TFL viruses from each infected individual with multiple lineages observed are indicated in S1 Table.

As with the SHEP-T2 panel described above, limiting the test panel to 12 viruses facilitates future screening of sera and monoclonal antibodies from large longitudinal vaccine trials. We used k-means clustering [38] to down-select a set of viruses that best represent the bnAb IC50 titer diversity found in the larger AMP placebo set. The process begins with 12 centroids, taken to be the neutralization sensitivity profiles of 12 randomly chosen viruses. For each iteration, the viruses are assigned to clusters, with the assignment of a given virus to the cluster whose centroid is closest, based on Euclidean distance in the space of Log IC50 titers (Methods), to the neutralization sensitivity profile of that virus. The centroid for each cluster is then re-computed as the geometric mean of each antibody response to the viruses assigned to the cluster, and the process is iterated until the clusters are stable. Thus, viruses with similar neutralization sensitivity profiles ultimately are grouped together. The most representative virus of a given cluster is the one whose neutralization sensitivity profile is closest to the centroid, and these 12 viruses were used to define REP-T2. The viral clusters and most central form of each cluster are shown in S6 Table, and the 12 panel representatives for REP-T2 are summarized in Fig 5.

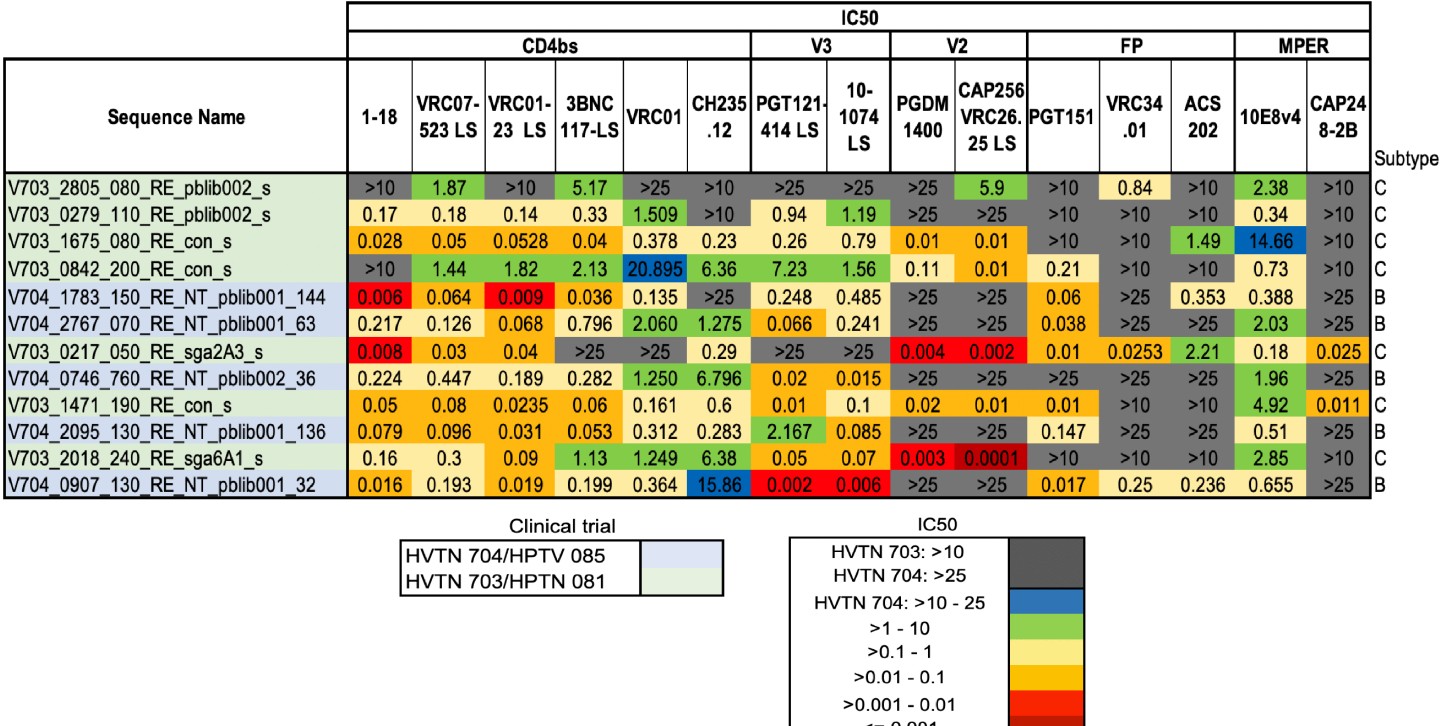

**Fig 5. IC50 scores for REP-T2 panel.** The color key represents the IC50 range in logarithmic scale. These viruses are selected to be representative of the spectrum of diversity of bnAb sensitivity profiles found among AMP placebo data and include just the centroids of 12 K means defined clusters, as described in the text. See S6 Table for the full clustering data.

### Tier phenotyping

To test the neutralization tier of the selected Envs, we tested the pseudoviruses bearing these Envs against 5 polyclonal IgG samples from HIV-positive individuals that had been previously down-selected from a large panel for having a suitable range of IC50 values. The 5 IgG samples were tested against pseudoviruses with known neutralization phenotypes to establish a reference range for Tiers 1A, 1B, 2, and 3. The tier of novel pseudoviruses was determined by comparing the geometric mean of IC50 values to the reference ranges. S7 Table includes the complete phenotyping data for SHEP-T2, the per-bnAb class sensitive panels, and for REP-T2, confirming that all newly chosen panel pseudoviruses (Figs 3–5) were either Tier 2 or 3, but not Tier 1. One pseudovirus that was very sensitive to V2 apex antibodies was classified as Tier 3. A closed Env conformation was confirmed by showing that none of the panel pseudoviruses were sensitive to neutralization by antibodies that depend on an open conformation for neutralization (e.g., F105 in S7 Table). Two of the viruses proposed for panels were not easily grown to enable sharing and repeated use as reference reagents, so those 2 were replaced by alternates that were readily expanded to enable sharing the reagent panels (see Fig 4).

## Discussion

### Heterologous breadth and depth

HIV-1 continues to evolve ever greater levels of genetic and antigenic diversity, resulting in increasing resistance to bnAbs at the population level [1–5]. A commonly used 208 virus panel of diverse HIV Env pseudoviruses [39] has been an invaluable resource to the field, but it has limitations, as these viruses were sampled during an older epoch in the HIV pandemic, between 1983 and 2008 (Fig 1A), when the circulating viruses were less divergent [1–3,5] (Fig 1C). Phylogenetic analyses of the global diversity of contemporary versus historic HIV Envs pseudovirus panels for antibody breadth and potency testing (Figs 1C, S1 and S2) highlight the need to update panels both for evaluating the potential value of both of bnAbs for prevention and therapeutic strategies, and even for detecting bnAb precursors as they arise and begin to undergo affinity maturation during vaccine trials and natural infection.

Some of these viruses are Tier 1 (Fig 1B) and therefore atypically neutralization sensitive and not representative of most circulating variants [11]. Finally, many were derived from Env sequences obtained during chronic infection and based on bulk PCR amplification [17,40]. In contrast, AMP pseudoviruses were sampled very early into the infection (i.e., good inferences of TFLs could be made), were relatively recent being isolated between 2017–2019 (Fig 3A), and were sequenced using methods that optimize accuracy and minimize *in vitro* recombination [17,40]. Thus, pseudoviruses from the 76 AMP placebo cases provide a good starting place for a more contemporary large Env panel. For SHEP-T2 and the antibody specific panels, we used the most sensitive pseudovirus from each individual when multiple TFL viruses were tested (S1 Table). We felt a better choice for a pseudovirus panel to assess breadth-potency at the population level, the REP-T2 panel, would be the least sensitive variant from each host, as the AMP trial demonstrated that the presence of bnAbs can select against more sensitive forms of the virus [20,37].

One critical, unresolved issue is that many globally important HIV-1 clades are still poorly represented among both the commonly used neutralizing panels and the AMP placebo viruses, and global shifts in clade distributions are also evident over time (Fig 6). Given that bnAbs often display clade specificity [34], this is an important issue. The AMP viruses are less representative of global clade and recombinant diversity than the 208 and 119 viral panels, as the AMP clinical trials were conducted in limited geographic regions where B, C and F clade viruses were dominant (Fig 1C). Furthermore, clade distributions are dynamic and shifting over time. CRF01, which was originally only common in southeast Asia, has been expanding in Europe and Australia [42]; Subtype A is increasingly sampled in Central Africa; CRF02 in West Africa; a distinctive A clade sublineage, A6, is common in eastern Europe [43], and the European epidemic has become much more complex in recent years [44]. Meanwhile, although the AMP placebo viruses provide a valuable set of relatively contemporary viruses for the globally most common C and B clades, the AMP samples are already between 6–8 years old (Fig 1A).

PLOS Pathogens

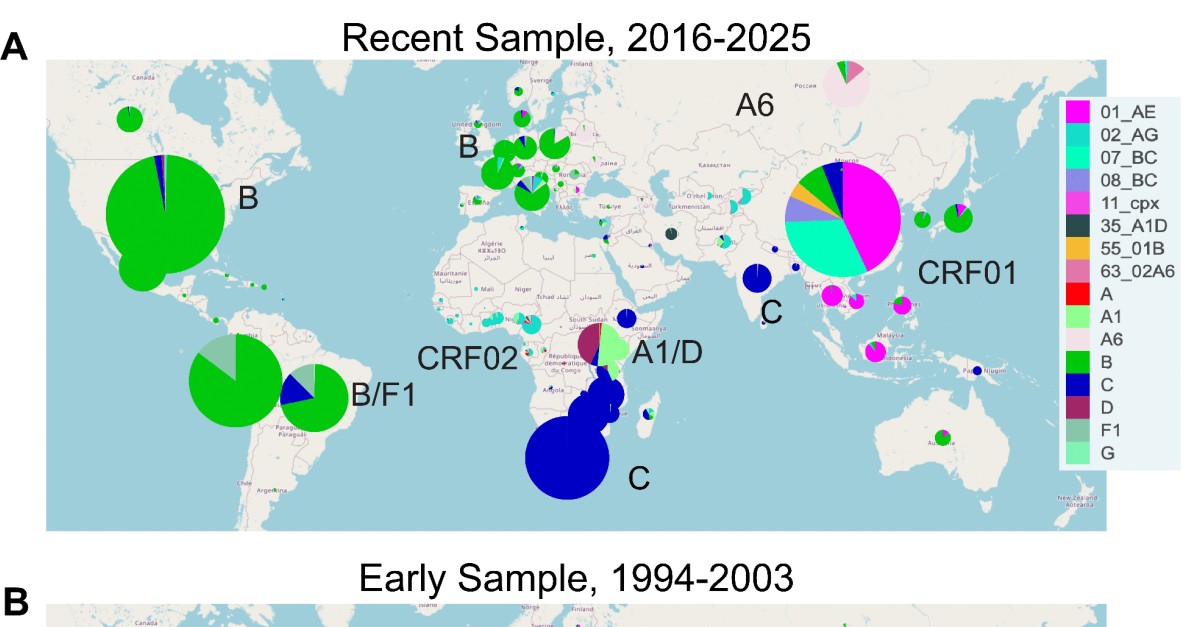

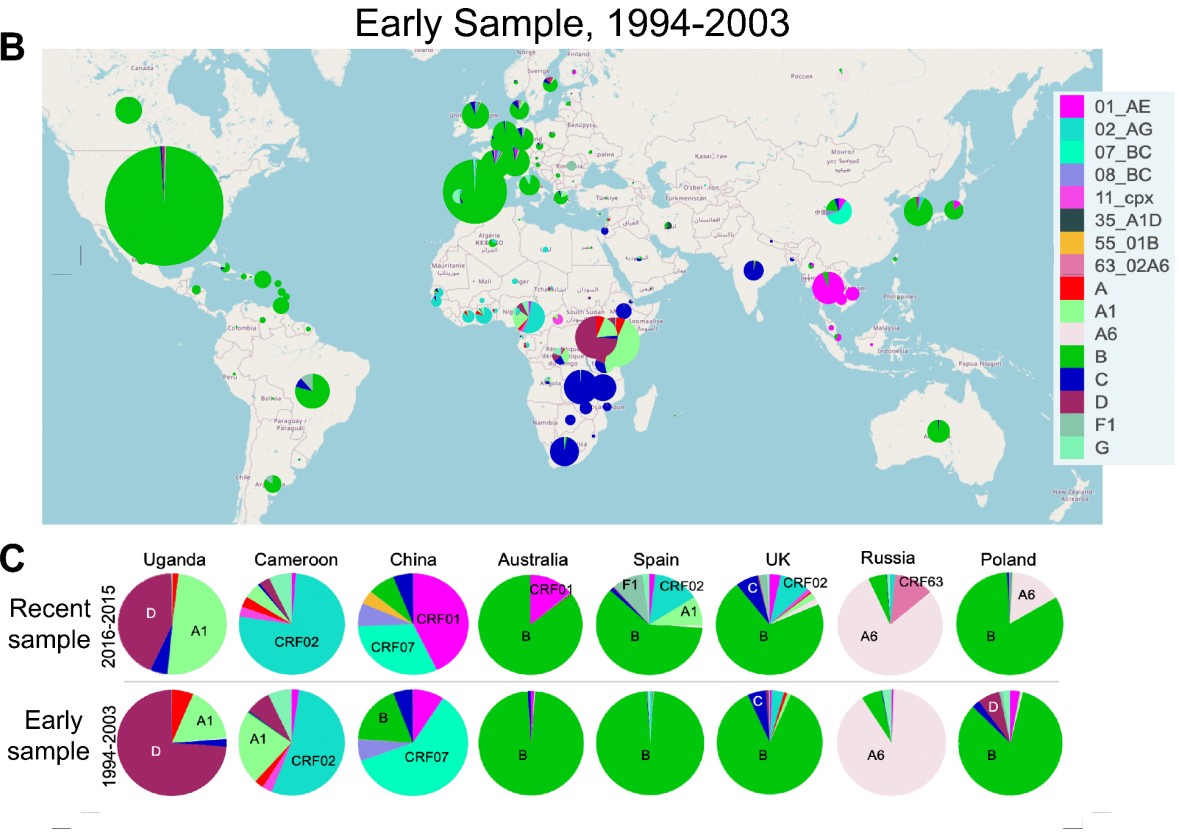

**Fig 6. Global distributions by country of the most common HIV-1 subtypes and circulating recombinant forms (CRFs), including those that have been sampled more than 1000 times each, in two 10 year windows. (A)** A global map of the distribution of common lineages during the most recent 10-year period 2016-2025. Common lineages in different global regions are highlighted, for example subtype B in North America (green) and C in southern African (blue), both of which have remained regionally dominant throughout the history of the HIV-1 pandemic [41]. **(B)** A global map of an earlier 10-year sampling period (1994-2023), illustrating regional transitions and overall increasing subtype diversity over time [41]. The lineages CRF63 (Russia), CRF01 (Southeast Aisa), CRF02 (West Africa), A6 (Eastern Europe), F1 (South America) are commonly sampled and increasing in both frequency and geographic spread but are under-represented in current pseudovirus panels (Fig 1). **(C)** A focused display of countries from the maps in (A) and (B) that exemplify where subtype distributions have shifted dramatically and are representative of regional trends. Subtype D used to be common

in East Africa, but it is now decreasingly sampled with subtype A1 increasing (e.g., Uganda). West Africa was historically very complex, but CRF02 is becoming increasing dominant (e.g., Cameroon). CRF01 is becoming more prevalent globally and is increasingly sampled in many countries (e.g., China and Australia) and beginning to have stronger presence in Europe (e.g., Spain and the United Kingdom (UK)). Russian sampling has been dominated by the A sublineage A6 for decades, and but A6 as begun to spread throughout eastern Europe and central Aisa (e.g., Poland) and CRF62, a CRF02/A6 recombinant form, has begun to be increasingly sampled in Russia. These maps were generated using the HIV LANL Database (https://www.hiv.lanl.gov/components/sequence/HIV/geo/geo.html) [41], and are constructed using Leaflet, opensource software https://www.hiv.lanl.gov/components/sequence/HIV/geo/geo.html BSD 2-Clause License Copyright 2010-2026, Volodymyr Agafonkin, which in turn uses OpenStreetMap https://www.open-streetmap.org/copyright, Creative Commons Attribution-ShareAlike 2.0 license (CC BY-SA 2.0, https://creativecommons.org/licenses/by-sa/2.0/).

Perhaps the most critical lesson to learn from the AMP placebo viruses is that it is important to continue to monitor contemporary viruses, and that these reagent panels based on AMP placebo viruses are placeholders. Carefully planned, deliberate, and consistent sampling would be highly beneficial at this time to create a more accurate portrait of global diversity, and this would in turn enable better neutralization panels to gauge the potential for antibody-based prevention and therapeutics to be effective across the globe, as well as to monitor and advance promising vaccine candidates.

## Materials and methods

### Phylogenetic trees and sequence alignments

Phylogenetic trees were generated from the amino acid alignment using the software FastTree 2.2 [45], using the JTT evolution model [46]. In the alignments used for the trees, the four hypervariable regions V1, V2, V4 and V5, which frequently evolve by insertion and deletion rather than base mutation and so are not readily aligned, were stripped from the env alignments, as were all positions that contained 90% or more gaps. Each tree includes 345 viral sequences from the full AMP trial (Figs 1C, S1 and S2) combined with a set of one of three commonly used and important virus panels. Unlike many of the sequences used for older panels, AMP sequences were generated using state of the art single-molecule, long read viral sequencing using the SMRT-UMI PacBio platform [18]. For AMP sequence selection, for each study participant 1–4 sequences were selected to best represent major founder lineages identified in early infection. When more than one lineage was present, sequences representing lineages found at a 10% or higher frequency were included [2]. We used the entire set of AMP sequences for the trees (Figs 1C, S1 and S2) to fully represent the diversity of the viruses sampled in the AMP study populations, but for other parts of this study we focused on just the AMP placebo participants so not to introduce a bias towards VRC01 resistant viruses in our new panel selection.

The three commonly used virus panels described in this paper were previously described and retrieved as follows:

- Doria-Rose et al. [6], 205 viral sequences from the 208 global viral panel commonly used to assess neutralizing breadth and potency. This panel was downloaded from the LANL HIV-1 database using the CATNAP tool accessed in July 2025 [7] (https://www.hiv.lanl.gov/components/sequence/HIV/neutralization/main.comp), in conjunction with the neutralizing antibody data from many of the studies that have historically used this reference set. The three viruses from the panel were not included in the phylogeny as the sequences were not available were: 271_11, T266_60, T33_7.

- Schoofs et al. [33], a smaller global panel of 119 viruses also commonly used to assess neutralizing breadth and potency. Eighty-one of these viruses were also found in the Doria Rose set, 38 were not. All viruses and accompanying neutralizing antibody data can be obtained via the CATNAP tool [7]. This panel is a slightly altered updated version of a panel first described by Seaman et al. [14].

- Rademeyer et al. [3], a panel of 200 C clade viral sequences also available in the CATNAP tool [7].

The 208 and 119 viral panels were multiclade and global, but Rademeyer et al., was comprised of only clade C viruses. Metadata including: the clade; year and country of isolation; overlaps between panel; and for AMP viruses, the clinical trial and treatment status, is provided in S8 Table. All neutralization data, aligned sequences, and virus panels are provided in

supplementary data tables and are also available at the Harvard Dataverse Repository: dataverse.harvard.edu/dataset.xhtml?persistentId = https://doi.org/10.7910/DVN/3AVJUQ. All env sequences in this study are available from GenBank, and accession numbers and metadata are listed in S8 Table. Additional information on this assay and supporting protocols may be found at: http://www.hiv.lanl.gov/content/nab-reference-strains/html/home.htm

### Neutralization assays

Neutralization sensitivity of the viruses assessed using the TZM-bl neutralization assay [47,48]. Neutralizing antibodies were measured as a function of reductions in luciferase (Luc) reporter gene expression after a single round of infection in TZM-bl cells as previously described [47,49]. The assay is formally optimized and validated [3] and was performed in compliance with Good Clinical Laboratory Practices [48], including participation in a formal proficiency testing program [50]. Additional information on this assay and supporting protocols may be found at: http://www.hiv.lanl.gov/content/nab-reference-strains/html/home.htm

Results used for the panel classification were expressed as IC50 values, which are the 50% inhibitory antibody concentration, reported in micrograms/mL (ug/ml), required to neutralize 50% of HIV-1 infection in TZM.bl target cells. Only IC50 data is shown in the panel figures as that was the data used to select the panels. This was because IC50 data has fewer undetected scores than IC80 data, and a broader distribution of values. However, IC80 values for all data are also provided in the S1 Table and were used to define a biologically meaningful threshold for Fig 2A and 2B, of IC80 < 1 ug/ml [20].

### Tier phenotyping reagents

Tier phenotyping was performed by using purified IgG fractions of serum samples from people living with HIV subtype C. Reference values were determined from assays with a panel of viruses having well characterized tier phenotypes; the samples are designated as SA-C10, SA-C48, SA-C72, SA-C74 and SA-C90. Additional tier phenotyping was performed by using monoclonal antibodies against epitopes that are typically exposed only on tier 1 viruses; these epitopes were probed with the V3-specific mAbs 2219 [50], 2557 [51], 3074 [52], 3869 [53], 447-52D [54] and 838-D [55], and the CD4bs mAbs 654-30D [56], 1008-30D, 729-30D [57] and F105 [58,59].

### Germline reverted antibody production

Germline reverted antibodies were produced in 293i cells by transient transfection, purified using Protein A chromatography, and filtered using 0.2um filtration (Duke Protein Production Facility). The protein was further purified with by FPLC size exclusion chromatography. They were stored in 25mM citric acid and 125mM sodium hydrochloride storage buffer (pH 6.0) at -80C.

### Comparing the AMP placebo virus with the 208 viral panel for the breadth and potency of neutralizing antibodies

The most relevant one-sequence-per-person AMP set of viruses for this purpose was used to select the REP-T2 viral panel, based on the most resistant TFL viruses were identified from each person being used to represent that person (S1 Table). For the 208 viral panel data, we used the summary statistics provided by the CATNAP [7] for the 13 antibodies with complete panel data available. Three of 13 of the antibodies tested against the AMP viruses (3BNC117, 10–1074, and CAP256 VRC26.25) were the LS version [29], with two mutations introduced into the Fc receptor to improve pharmacokinetics were used for this study, whereas for the 208 panel complete data was only available for the non-LS version of these antibodies; as these mutations are unlikely to impact neutralization potency we included them. Also, the IC50 and IC80 scores provided in CATNAP are a composite score from the many papers that have tested a given antibody/virus combination, with the geometric mean score averaged across all published studies was used. To standardize a across studies we tallied the number of viruses that had IC80 of <1 ug/ml [20] in Fig 2A and 2B. To statistically evaluate the null

hypothesis that the 13 bnAbs tested had equivalent capacities to neutralize viruses with an IC80 of <1ug/ml in the 208 panel relative to the AMP placebo viruses a Wilcoxon signed-rank test was used, implemented using the python scipy. stats package function wilcoxon with the keywords "alternative=two-sided" and "method=exact".

**K-means clustering**

Clustering was performed using the k-means algorithm as implemented in the scikit-learn package [38] as scklearn.cluster.KMeans. Each row of the heat map is assigned a 15-dimensional vector corresponding to the values in that row of the heat map shown in Fig 5; this vector is what we are calling the neutralization sensitivity profile of a given virus. Antibodies were tested to different dilutions in different laboratories, and different amount of antibody was available for testing, so in some cases the threshold of detection was > 10, in others >25, and in a few cases >100. K = 12 centroids are computed in this 15-dimensional space using the logarithm of the IC50 values, and for each centroid, rows are identified that "belong" to that centroid -- a row belongs to a centroid if that centroid is the closest (of the K = 12) centroid to the vector associated with that row. Also, for each centroid, the row that is closest to the centroid is identified as the representative natural virus for that cluster. Note that it is possible for a cluster to contain only a single member (which would then be the cluster's representative). Note that there are several single-element clusters (S6 Table); this is not surprisingly they capture particularly resistant and sensitive variants.

**Supporting information**

**S1 Fig. Midpoint rooted phylogenetic tree comparing a global panel of 119 viruses used to assess bnAb breadth and potency [30–33] to 154 AMP V703 and 191 V704 trial sequences [2,17].** Phylogenetic analysis shows that within B and C clades the 119 viral panel sequences are not representative of the diversity sampled in the more contemporary AMP trial. Note that the exact composition of the panel can vary by a few viruses between studies. The phylogenetic tree was based on data from a version of the 119 virus panel that was downloaded using the CATNAP tool from the Los Alamos Database accessed on July 1, 2025 [7].
(PDF)

**S2 Fig. Midpoint rooted phylogenetic tree comparing a global panel of 200 clade C viruses used to assess bnAb breadth and potency [30–33] to 154 AMP V703 and 191 V704 trial sequences [2,17].** Of all commonly used panels described in this paper, this exclusively clade C panel, while lacking representation for all other clades, was the closest to contemporary C clade viruses sampled from the AMP V703 participants. Notably though, within the clade C subtree, AMP sequences tend to have longer branch lengths than the clade C panel pseudoviruses and are more spread out in the tree. The phylogenetic tree was based on data from a version of the 200 virus panel that was downloaded using the CATNAP tool from the Los Alamos Database accessed on July 1, 2025 [7].
(PDF)

**S3 Fig. Details regarding V2 apex bnAb signatures within the V2 apex bnAb sensitive panel for detection of developing responses.** (A) The frequency of V2 apex bnAb sensitivity/resistance signatures as defined in Bricault et al. [34] in the class specific panel versus other AMP viruses. The height of the letter is indicative of the frequency of the amino acid in a given position in each group. An O indicates an N-linked glycosylation site. Blue are sensitivity signatures, red are resistance signatures, and black amino acids were not significantly associated with either one. (B) On the left we illustrate (also see S5 Table) the enhanced sensitivity of C clade and F/BF viruses relative to B clade viruses among the AMP collection of viruses to neutralization by the V2 apex bnAb CAP256 VRC26.25 LS (far left) and PGDM1400 (second to the left). On the right we show the distribution of hypervariable region characteristics that were associated with V2 apex bnAb neutralization [34]. As B clade viruses tend to be resistant regardless of loop characteristics, we restricted these comparisons to AMP C clade viruses. Positive V2 charge and shorter combined V1V2 hypervariable region lengths are signatures

of sensitivity to V2 Apex antibodies [34], and V2 hypervariable loop positive charge and shorter hypervariable regions are slightly enriched in the sensitive panel. (C) The sequence alignment for the V2 apex bnAb sensitive panel across the epitope region.
(PDF)

**S4 Fig. Details regarding V3 glycan bnAb signatures within the V3 glycan bnAb sensitive panel for detection of developing responses.** (A) The frequency of V3 glycan bnAb sensitivity/resistance signatures as defined in Bricault et al. [34] in the class specific panel versus other AMP viruses. The height of the letter is indicative of the frequency of the amino acid in a given position in each group. An O indicates an N-linked glycosylation site. Blue are sensitivity signatures, red are resistance signatures, and black amino acids were not significantly associated with either one. (B) Shorter V1 hypervariable length and shorter combined V1V2 hypervariable region lengths are associated with enhanced sensitivity to V3 glycan bnAbs [34], and there is an enrichment for shorter V1 and V1 + V2 hypervariable regions in the V3 glycan sensitivity panel. (C) The sequence alignment for the V3 glycan bnAb sensitive panel across the epitope region.
(PDF)

**S5 Fig. Details regarding CD4bs bnAb signatures within the CD4 bnAb sensitive panel for detection of developing responses.** (A) The frequency of CD4bs bnAb sensitivity/resistance signatures as defined in Bricault et al. [34] in the class specific panel versus other AMP viruses. The height of the letter is indicative of the frequency of the amino acid in a given position in each group. An O indicates an N-linked glycosylation site. Blue are sensitivity signatures, red are resistance signatures, and black amino acids were not significantly associated with either one. (B) Shorter V1 + V2 region lengths with fewer glycans were associated with CD4bs bnAb sensitivity [34], and while shorter combined V1V2 lengths were not enriched in the sensitive panel, fewer glycans in the V1V2 loop regions were. Shorter V5 loops with fewer glycans were also previously associated with enhanced CD4bs bnAb sensitivity [34], and both of these characteristics were enriched in the sensitive panel. (C) The sequence alignment for the CD4bs bnAb sensitive panel across the epitope region.
(PDF)

**S6 Fig. Details regarding MPER bnAb signatures within the MPER sensitive panel for detection of developing responses.** (A) The frequency of MPER sensitivity/resistance signatures as defined in Bricault et al. [34] in the class specific panel versus other AMP viruses. The height of the letter is indicative of the frequency of the amino acid in a given position in each group. An O indicates an N-linked glycosylation site. Blue are sensitivity signatures, red are resistance signatures, and black amino acids were not significantly associated with either one. (B) Shorter V1 region lengths with fewer glycans were associated with MPER sensitivity [34], and these were not enriched in the MPER sensitive panel. (C) The sequence alignment for the CD4bs bnAb sensitive panel across the epitope region. This a highly conserved epitope.
(PDF)

**S1 Table. IC50 and IC80 scores for the AMP V703 and V704 study sequences.** (A) IC50 AMP V704, (B) IC50 AMP V703, (C) IC80 AMP V704, and (D) IC80 AMP V703.When more than one transmitted founder lineage was found in an individual, the set of TFLs are grouped in these tables. For designing SHEP-T2 and class specific sensitive panels, we used the most sensitive variant overall to represent an individual, and those are marked in grey on the left. For the REP-T2 the least sensitive from an individual was used, are these are marked in yellow. "ND" means that the assay was not done.
(XLSX)

**S2 Table. Nomenclature key and critical sequence metadata.** (A) V703 study sequences. (B) V704 study sequences.
(XLSX)

**S3 Table. Ranking of the AMP viruses for creating the SHEP-T2 bnAb sensitive screening panel.** Note that only 13 of the 15 antibodies tested (see S1 Table) were included here. This is because the full neutralization data for the antibody VRC01–23 LS was not available at the time this panel was first designed and pseudovirus evaluation was initiated, and CAP248-2B AMP data indicated that it was not a broadly neutralizing antibody, so both were excluded it from consideration. We began the work to characterize this panel and to share the panel reagents prior to the complete VRC01–23 LS data being obtained, but VRC01–23 LS data is included in the SHEP-T2 panel data and is shown in Fig 3. Fortunately, and as expected, the 12 viruses selected for the panel were also very sensitive to VRC01–23 LS, and the two viruses most sensitive to VRC01–23 LS viruses were already included in the panel. The IC50 mean rank orders the viruses according to their overall sensitivity to this panel of antibodies. In addition, the 3 lowest antibody scores for each antibody are written in white and underlined, and we required that at least 2 of these 3 were included the SHEP-T2 panel, which is why two of the lower ranked viruses (ranks 8 and 11) were replaced with viruses that were ranked higher and less sensitive overall (ranks 17 and 25), as these viruses were particularly sensitive to the V2 apex antibodies and to FP antibody ACS202.
(XLSX)

**S4 Table. Twenty six germline precursors antibodies that ultimately give rise to bnAbs show no IC50 against any of the 40 viruses included in the SHEP-T2 and class-specific antibody panels.** (A) Virus details and panel inclusion. (B) IC50 and IC80 responses for each of the 26 antibodies tested; there was no detectable activity against any of the AMP viruses.
(XLSX)

**S5 Table. Clade specificity of V2 apex antibodies.** IC50s of V2 apex antibodies against: (A) B clade AMP viruses, (B) C clade AMP viruses, or (C) other subtypes and recombinant viruses found among the AMP placebo HIV-1 infections. The V2 apex bNab CAP256 VRC26.25 LS has substantial breadth and great potency among C clade viruses but has almost no neutralizing activity against B clade viruses. PGDM1400 shares this pattern but is less extreme, with a few modest B clade responses and less potent but still strong neutralization of C clade viruses. F clade viruses (in Table C) can be sensitivity to both antibodies and are generally more like C clade viruses than B clade. B/F recombinants, indicated by the forward slash, were each checked for recombination breakpoints, and all were found to be F-like in the relevant V2 apex epitope region.
(XLSX)

**S6 Table. K-means clusters used to define the REP-T2 panel.** (A) The REP-T2 panel. (B). The centroids of each cluster were used as panel members and are marked in pink.
(XLSX)

**S7 Table. Establishing the Tier status by using 5 standardized sera and a set of monoclonals that are typically only exposed on Tier 1 viruses in open conformation.** The standardized sera are called SA-C10, SA-C48, SA-C72, SA-C74, and SA-C90. Members of the SHEP-T2 panel were sometimes included in the sensitive panels for a particular class. *The two viruses with a single asterisk did not grow well, so were not practical for use in panels were replaced with the viruses with the marked with ** so reagent stocks could be generated and shared.
(XLSX)

**S8 Table. Metadata associated with HIV-1 pseudovirus sequence data used in this study.** This table includes the available metadata associated used with the pseudovirus sequences used here. It includes metadata for 205 of the 208 viruses with available data representing the global panel (Fig 1C), the 119 virus more limited global panel (S1 Fig), the 200 virus C clade panel (S2 Fig), and AMP V703 and V704 sequences. The metadata includes sequence names,

GenBank accession numbers, AMP identifiers and treatment status, viral clade or circulating recombinant form designations, the country of sampling, and the year of isolation.
(XLSX)

## Acknowledgments

We thank the team at the Los Alamos HIV database for their dedicated work on the CATNAP tool help, and their support with using the data they curate and their analyses tools: Elizabeth-Sharon Damayanthi Fung, Hyejin Yoon, Jennifer Macke, and Jennifer Mamrosh. We also thank Dan Barouch, PI of one of the grants funding this work.

## Author contributions

**Conceptualization:** Bette Korber, Nicole A. Doria-Rose, David Montefiori, Elena E. Giorgi.

**Data curation:** Kelli Greene, Hongmei Gao, Elizabeth Domin.

**Formal analysis:** Bette Korber, James Theiler, Kshitij Wagh, Elena E. Giorgi.

**Funding acquisition:** Kelli Greene, Hongmei Gao, Penny L Moore, Carolyn Williamson, James I. Mullins, David Montefiori.

**Investigation:** Michael S. Seaman, Nonhlanhla N. Mkhize, Kelli Greene, Hongmei Gao, Xiaoying Shen, Elizabeth Domin, Haili Tang, Penny L Moore, David Montefiori.

**Methodology:** Bette Korber, James Theiler.

**Project administration:** Kelli Greene, Hongmei Gao.

**Resources:** Michael S. Seaman, Nonhlanhla N. Mkhize, Penny L Moore, Carolyn Williamson, James I. Mullins, Nicole A. Doria-Rose, David Montefiori.

**Software:** Bette Korber, James Theiler, Kshitij Wagh.

**Supervision:** Bette Korber, Michael S. Seaman, Kelli Greene, David Montefiori.

**Validation:** Xiaoying Shen, Elizabeth Domin, Haili Tang.

**Visualization:** Bette Korber, David Montefiori.

**Writing – original draft:** Bette Korber, Elena E. Giorgi.

**Writing – review & editing:** Bette Korber, Michael S. Seaman, Nonhlanhla N. Mkhize, Kelli Greene, Xiaoying Shen, James Theiler, Kshitij Wagh, Penny L Moore, Carolyn Williamson, James I. Mullins, Nicole A. Doria-Rose, David Montefiori, Elena E. Giorgi.

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
