## [Decision Letter · Decision Letter 0]

23 Dec 2025

PPATHOGENS-D-25-02897

Contemporary HIV-1 envelope pseudovirus panels for detecting and assessing B cell lineages with broadly neutralizing antibody potential

PLOS Pathogens

Dear Dr. Giorgi,

Thank you for submitting your manuscript to PLOS Pathogens. After careful consideration, we feel that it has merit but does not fully meet PLOS Pathogens's publication criteria as it currently stands. Therefore, we invite you to submit a revised version of the manuscript that addresses the points raised during the review process.

We look forward to receiving your revised manuscript.

Kind regards,

Guido Silvestri

Academic Editor

PLOS Pathogens

Susan Ross

Section Editor

PLOS Pathogens

Sumita Bhaduri-McIntosh

Editor-in-Chief

PLOS Pathogens

orcid.org/0000-0003-2946-9497

Michael Malim

Editor-in-Chief

PLOS Pathogens

orcid.org/0000-0002-7699-2064

**Journal Requirements:**

At this stage, the following Authors/Authors require contributions: Bette Korber, Michael S. Seaman, Nonhlanhla N. Mkhize, Kelli Greene, Hongmei Gao, Xiaoying Shen, Elizabeth Domin, Haili Tang, James Theiler, Kshitij Wagh, Penny L Moore, Carolyn Williamson, James I. Mullins, Nicole A. Doria-Rose, David Montefiori, and Elena Giorgi. Please ensure that the full contributions of each author are acknowledged in the "Add/Edit/Remove Authors" section of our submission form.

Potential Copyright Issues:

- Figure 6. Please (a) provide a direct link to the base layer of the map (i.e., the country or region border shape) and ensure this is also included in the figure legend; and (b) provide a link to the terms of use / license information for the base layer image or shapefile. We cannot publish proprietary or copyrighted maps (e.g. Google Maps, Mapquest) and the terms of use for your map base layer must be compatible with our CC BY 4.0 license.

**Reviewers' Comments:**

Reviewer's Responses to Questions

**Part I - Summary**

Reviewer #1: Success of HIV-1 vaccinations in pre-clinical and clinical trials is assessed using stringent Tier-2 HIV-1 virus neutralizing assays that tend to represent global diversity of circulating strains. However, current panels of viruses used in standardized assays were derived from strains/envelopes harvested almost 2 decades ago.

Here, Betty Korber et al, report the evolution of HIV-1 circulating strains in regards with global diversity of Envelopes, recombinant forms and hence, reduced value in assessing impact of vaccine induced immunity. Using transmitted founder virus strains from an Antibody Mediated Protection (AMP) trials, HVTN 703 and 704, the authors show the value of updated pseudovirus panels that can highlight improved representation of contemporary virus strains/panels that need to be periodically updated to support vaccine studies or HIV-1 vaccine outcomes in general.

The manuscript is very well written, and the data is compelling that designing new panels to quantify HIV-1 neutralizing capability in vitro, such as with the SHEP-T2 and the REP-T2 can substantially support HIV-1 vaccine trials as well as understanding elicitation of early breadth using germline targeted and shepherding approaches.

I Recommend publication of this exciting advance in assessing HIV-1 vaccination trials.

I only have one minor comment in the supplementary tables.

Minor Comments.

1. The headers needed for the tables in Table S1A is only included in Table S1B (IC50, and targeted Env regions such as CD4bs, V3, V2,FP, MPER)

Reviewer #2: Korber et al have conducted an insightful follow-up study building on previous research regarding the efficacy trials of Antibody Mediated Prevention (AMP). They underscore the importance of refreshing the reference panel of HIV-1 Envelope pseudotyped viruses that are routinely used to evaluate broadly neutralizing antibodies (bNab), as well as to assess the quality of antibody responses prompted by vaccines. Given the passage of time since the original panel was established and the ongoing evolution of HIV-1, this study is both timely and necessary.

Panels that incorporate current and recently transmitted HIV-1 strains have the potential to provide greater insights into bNab development and improve the design of therapeutic regimens. By utilizing envelopes from participants who received placebos in the AMP efficacy trials, the authors have developed panels that can detect early neutralizing responses and class-specific broadly neutralizing antibodies, along with antibody responses to contemporary and naturally circulating variants. While these panels may not be perfect (lacking representation from all global clades), they offer valuable tools for investigating bNab in various contexts.

Thus, this study is significant and will help advance the field of broadly neutralizing antibodies for the prevention and treatment of HIV-1. The methodology applied mirrors that of the previously established reference panel, originally developed from strains collected between 1998 and 2010, which served as a comparator for the new panels created. Therefore, there are no major issues with this study. However, below are some minor points to address for clarity and consistency.

**Part II – Major Issues: Key Experiments Required for Acceptance**

Please use this section to detail the key new experiments or modifications of existing experiments that should be absolutely required to validate study conclusions.required to validate study conclusions.required to validate study conclusions.required to validate study conclusions.

Reviewer #1: None except spell checks through the manuscript.

Reviewer #2: None

**Part III – Minor Issues: Editorial and Data Presentation Modifications**

Reviewer #1: Minor Comments.

1. The headers needed for the tables in Table S1A is only included in Table S1B (IC50, and targeted Env regions such as CD4bs, V3, V2,FP, MPER)

Reviewer #2: Another thorough read

1.- Extra words

- Line 388 in Discussion, “and for even for detecting,” one of the “for” should be deleted.

- Line 562 in Figure legend 1, “longer branch lengths within both and B and C clades,” one of the “and” should be deleted.

2.- Mostly in the Discussion and Materials and Methods, there is inconsistent formatting for figures and tables. Sometimes bold, sometimes not.

- Lines 386, 387, 395, 411, 417, 480, 481, 500, and 510.

Figure legends and Supplemental tables

1.- Main figure legends have a lot of discussion of the results instead of description of the figure/analysis, thus there are similar sentences in the Results section and Figure legends, making the paper very repetitive.

2.- The Supplemental Tables are not described chronologically. The authors start with Table S2, then go to Table S5, then Tables S1, S3 and S4. In addition, Table S8 is mentioned several times and there is no Table S8 in the manuscript.

PLOS authors have the option to publish the peer review history of their article (what does this mean?). If published, this will include your full peer review and any attached files.). If published, this will include your full peer review and any attached files.). If published, this will include your full peer review and any attached files.). If published, this will include your full peer review and any attached files.

...

Reviewer #1: No

Reviewer #2: No

**Figure resubmission:**
---

## [Editor Report · Decision Letter 1]

24 Feb 2026

Dear Dr Giorgi,

We are pleased to inform you that your manuscript 'Contemporary HIV-1 envelope pseudovirus panels for detecting and assessing B cell lineages with broadly neutralizing antibody potential' has been provisionally accepted for publication in PLOS Pathogens.

Best regards,

Guido Silvestri

Academic Editor

PLOS Pathogens

Susan Ross

Section Editor

PLOS Pathogens

Sumita Bhaduri-McIntosh

Editor-in-Chief

PLOS Pathogens

orcid.org/0000-0003-2946-9497

Michael Malim

Editor-in-Chief

PLOS Pathogens

orcid.org/0000-0002-7699-2064
---

## [Editor Report · Acceptance letter]

Dear Dr Giorgi,

We are delighted to inform you that your manuscript, "Contemporary HIV-1 envelope pseudovirus panels for detecting and assessing B cell lineages with broadly neutralizing antibody potential," has been formally accepted for publication in PLOS Pathogens.

Best regards,

Sumita Bhaduri-McIntosh

Editor-in-Chief

PLOS Pathogens

orcid.org/0000-0003-2946-9497

Michael Malim

Editor-in-Chief

PLOS Pathogens

orcid.org/0000-0002-7699-2064